# The Common Factors of Grit, Hope, and Optimism Differentially Influence Suicide Resilience

**DOI:** 10.3390/ijerph17249588

**Published:** 2020-12-21

**Authors:** Déjà N. Clement, LaRicka R. Wingate, Ashley B. Cole, Victoria M. O’Keefe, David W. Hollingsworth, Collin L. Davidson, Jameson K. Hirsch

**Affiliations:** 1Department of Psychology, Oklahoma State University, Stillwater, OK 74075, USA; dejclem@okstate.edu (D.N.C.); abcole@okstate.edu (A.B.C.); 2Bloomberg School of Public Health, John Hopkins University, Baltimore, MD 21218, USA; vokeefe3@jhu.edu; 3Tuscaloosa Veterans Affair Medical Center, Tuscaloosa, AL 35404, USA; dhollingsworth@gmail.com; 4Allina Health, Minneapolis, MN 55407, USA; Collin.Davidson@allina.com; 5Department of Psychology, East Tennessee State University, Johnson City, TN 37614, USA; Hirsch@mail.etsu.edu

**Keywords:** grit, hope, hopelessness, hopelessness, suicide ideation

## Abstract

No study to date has simultaneously examined the commonalities and unique aspects of positive psychological factors and whether these factors uniquely account for a reduction in suicide risk. Using a factor analytic approach, the current study examined the relationships between grit, hope, optimism, and their unique and overlapping relationships in predicting suicide ideation. Results of principle axis factor analysis demonstrated close relationships between these variables at both the construct and item level. Item-level analyses supported a five-factor solution *(Stick-to-Itiveness, Poor Future, Consistency of Interest, Positive Future, and Poor Pathways)*. Four of the five factors (excluding *Stick-to-Itiveness)* were associated with suicide ideation. Additionally, results of a multiple regression analysis indicated that two of the five factors (*Consistency of Interest* and *Positive Future*) negatively predicted suicide ideation while *Poor Future* positively predicted suicide ideation. Implications regarding the interrelationships between grit, hope, and optimism with suicide ideation are discussed.

## 1. Introduction

Suicide is a complex phenomenon that presents as a public health concern worldwide. Suicide phenomenon may be understood through various social, political, cultural, and economic factors (i.e., poverty, personality traits, coping mechanisms, and environmental health) [1]. Recent data indicate that suicide deaths in the US have surged to the highest levels in nearly 30 years [2]. Suicide is currently the second leading cause of death for young adults of ages 15–34 in the US [3]. Empirical research has identified hundreds of suicide risk factors [4]; however, these factors may have limitations for clinical utility [5]. Counter to the extant suicide risk-focused approach, some researchers have suggested a potential solution to better understanding suicide may be through a positive psychological lens [6,7,8,9]. Taking a positive psychological approach to examining suicide is defined as examining the positive emotions, thinking patterns, and experiences that decrease suicide ideation and behavior [8]. Theoretically, the presence of protective factors may indicate that an individual is at lower risk for attempting and/or dying by suicide in their lifetime by buffering against suicide risk factors and increasing suicide resilience. Suicide resilience is defined as “the perceived ability, resources, or competence to regulate suicide-related thoughts, feelings, and attitudes [9]. While many factors (e.g., social connectedness, extroversion, and reasons for living) have previously been identified as negatively related to suicide ideation and behaviors [10,11,12,13,14], limited research has examined multiple positive psychological factors simultaneously and their unique predictive validity in relation to suicide risk.

Studies of grit, hope, and optimism have demonstrated that these future oriented and goal-related positive psychology factors are independently negatively associated with suicide ideation (e.g., [14,15,16,17,18,19]). It is currently unknown whether there are common features among these positive psychology variables that account for their protective nature or isolated components that are uniquely protective and responsible for reducing suicide risk. These variables warrant further study in efforts to parse distinct qualities that account for possible increase in suicide resilience. If the common features shared across these positive psychological variables can be identified, findings could have important implications for suicide interventions.

Grit is a future-oriented, goal-related positive psychological construct that has received recent empirical support as a protective factor against suicide. Grit is defined as an intrapersonal psychological strength characterized by passion and ability to pursue long-term goals and a willingness to persevere through barriers that prevent goal attainment [16,20]. Grit also encompasses working strongly toward challenges and maintaining interest and effort over time despite adversity and failure [20]. Previous research has demonstrated that grit was positively associated with positive affect, happiness, and life satisfaction, and grit was negatively associated with negative affect [21,22]. Grit, along with optimism, forgiveness, and meaning in life, were found to be conceptually related to high levels of spirituality; these collective constructs were also inversely related to symptoms of depression [23]. While this protective factor is a relatively new construct in the suicidology literature, it has been shown to enhance meaning in life while reducing suicide ideation [16]. Further, grit has been demonstrated to significantly moderate and mediate the relationship between depression and suicide ideation in a sample of Korean adults [24]. Additionally, grit has been demonstrated to significantly moderate the relationships between hopelessness and current suicide ideation, and between hopelessness and resolved plans and preparations for suicide [25]. While hope and grit share conceptual similarities and empirical findings have demonstrated that they each act as resiliency factors that buffer the risk for suicide, no research exists on the relationship between these two protective factors and their relationship to suicide.

Similar to grit, dispositional hope is a future-oriented positive psychology concept that has received empirical support as a protective factor against suicide. According to Hope Theory, hope encompasses: (1) goal setting; (2) pathways or perceived ability to develop plausible routes of achieving goals; and (3) agency or the motivation to achieve desired goals influenced by self-perceptions about one’s ability to attain goals in the past, present, and future [26,27,28]. Studies have found that hope and its three components negatively predict various indicators of suicide risk, including symptoms of depression, rumination, and suicide ideation [29]. Although hope is negatively associated with certain suicide risk factors, it has been positively related to an increased capability for enacting suicide in more than one sample [10,30]. It is evident that hope has protective qualities with respect to suicide risk; however, the positive association with increased acquired capability suggests a possible bidirectional effect. Therefore, future research is needed to examine the specific qualities of hope to better understand the nature of hope and its relationship with suicide ideation.

In addition to hope, hopelessness has been defined as holding negative expectancies about future life outcomes [31]. Although hopelessness is not considered to be a positive psychological concept, it is relevant to the present discussion because scholars have studied low hopelessness (low scores on Beck’s Hopelessness Scale) as a proxy for “hope” [31,32,33]. This is different from Snyder’s Hope Theory, in that individuals who are low in hopelessness may not exhibit goal-oriented behaviors or have positive expectancies for life. In one study of individuals diagnosed with schizophrenia, those with higher levels of insecure attachment exhibited lower levels of hope (more hopelessness [32]). Another study of patients after discharge from the hospital indicated that levels of hope might be increased among patients receiving peer support [33]. Additional research is needed to better understand the relationship of low hopelessness to suicide ideation.

Dispositional optimism is defined as the general belief that one’s future will be positive and favorable [34,35]. Research has demonstrated that dispositional optimism is related to lower incidences of suicide ideation and behavior [8,14,35,36]. Optimism has also been found to moderate the relationship between psychological correlates of ideation, such as rumination, hopelessness, thwarted belongingness, and perceived burdensomeness, with suicide ideation [8,29,30,37,38]. Past research has demonstrated moderate to large correlations between hopelessness and dispositional optimism [13,38], indicating that these constructs are related yet distinct. Similarly, Steed (2001) found moderate to large correlations between hopelessness, optimism, and hope, as well as a relationship between these variables and negative affect and perceived stress. Findings also revealed that hopelessness, optimism, and hope were similar in their convergent and discriminant validity. Currently, the Steed (2001) study is the only known study to examine these three constructs together.

Surprisingly, only a handful of studies have examined the constructs of grit, hope, and optimism in any arrangement or combination [25,29,30,39,40,41,42], and to our knowledge, no research has simultaneously examined the similarities of these constructs and their relationship with suicide ideation. The aims of the current study were to: (1) better understand the nuanced relationships between grit, hope, low hopelessness, and optimism; (2) identify common features of these variables that account for their protective nature; and (3) uncover whether there are distinct components that uniquely protect against suicide ideation. In line with previous findings [25,30,40], it was hypothesized that all positive psychological variables (optimism, hope, low hopelessness, and grit) would be positively related to each other, and when examined in a regression analysis, would be differentially predictive of suicide ideation. Factor analytic techniques were used to examine relationships between the positive psychological variables at both the construct and item levels. Given that the factor analyses were exploratory in nature, no specific factor analytic hypotheses were proffered. Additionally, the correlational relationships between the extracted factors and suicide ideation were explored to determine whether these extracted factors may account for potential reduction in suicide risk. Again, because the analysis was exploratory in nature, no specific hypotheses were proffered. Increased research is needed to better understand whether one factor is comparatively more salient in the context of suicide. This additional knowledge could have implications for both understanding their mechanisms of influence, and how they specifically function as protective factors.

## 2. Materials and Methods

### 2.1. Statistical Methods

A two-tailed bivariate correlation analysis was used to assess the relationship between the variables. Factor analytic techniques were conducted to examine relationships between the variables at both the construct and item levels. Additionally, correlational analyses of the extracted factors and suicide ideation were explored.

### 2.2. Participants

Participants included a convenience sample of 542 college students, at a large Midwestern university that opted into an online research participant’s pool. Participants were not taught any class information regarding study content in class. Ages of the participants ranged from 18 to 50 (*M* = 20.10, SD = 8.82). Participants were predominately female (76.9%). A total of 83% of the participants identified as Caucasian, 7% as American Indian, 5.7% as Hispanic/Latino, 3.7% as African American/Black, 2.6% as Biracial, 2.2% as Asian/Asian-American, and 1.3% as other.

### 2.3. Materials

Participants completed self-report questionnaires online through a research and data management system (SONA). Some participants received negligible course points in exchange for their participation. The university Institutional Review Board (IRB) approved all study procedures, and all participants provided informed consent prior to completing any of the questionnaires.

Demographics Questionnaire. Demographic information included questions about ethnicity/race, age, and sex.

Revised Life Orientation Test (LOT-R). The LOT-R [43] is a 10-item self-report questionnaire that measures levels of optimism. All items are scored on a 5-point Likert-type scale with response options ranging from 0 (*strongly disagree*) to 4 (*strongly agree*). In previous research within a similar sample, the LOT-R demonstrated good internal consistency (*α* = 0.85) [40]. The LOT-R demonstrated good internal consistency (*α* = 0.80) in the present study.

The Grit Scale. The Grit Scale [21] is a 12-item self-report measure that assesses levels of grit or persistence to work towards long-term goals. All items are scored on a 5-point Likert-type scale ranging from 1 (*not at all like me*) to 5 (*very much like me*). The Grit Scale contains two subscales, Consistency of Interest and Perseverance of Effort, which are comprised of 6 items each. The Grit Scale demonstrated good internal consistency (*α* = 0.81) in the current study. In a study that examined suicide risk and grit among college students, the Grit Scale demonstrated acceptable internal consistency (*α* = 0.79) [44].

Revised Trait Hope Scale (HS-R2). The HS-R2 [45] is an 18-item self-report questionnaire that measures the construct of hope and its three subscales: goals, pathways, and agency. All items are scored on an 8-point Likert-type scale ranging from 1 (definitely false) to 8 (definitely true). In the present study, the HS-R2 demonstrated excellent internal consistency (*α* = 0.90), and each subscale demonstrated adequate internal consistency (goals: *α* = 0.78, pathways: *α* = 0.73, agency: *α* = 0.82) in the current study. In a study that examined the Hope Scale and its three subscales in a sample of college students, the subscales’ alpha coefficients were 0.90 (goals), 0.81 (pathways), and 0.89 (agency) [46].

Beck’s Hopelessness Scale (BHS). The BHS [47] is a 20-item self-report questionnaire that assesses levels of hopelessness. The original BHS uses a dichotomous (yes or no) response format. In line with previous empirical recommendations [48], the current study used the adapted BHS, which includes a 5-point Likert-type scale ranging from 1 (*strongly disagree*) to 5 (*strongly agree*). The BHS was reverse scored to examine low hopelessness, or high levels of hope [32,33]. In a previous study that examined suicide risk with hopelessness in an undergraduate college sample, the BHS demonstrated excellent reliability (*α* = 0.93). The BHS demonstrated excellent internal consistency (*α* = 0.91) in the current study.

Hopelessness-Depression Symptom Questionnaire—Suicidality Subscale (HDSQ-SS). The HDSQ-SS [49] is a 4-item subscale that measures suicide risk. Items are scored on a scale ranging from 0 to 3, with varied responses corresponding to each number depending on the item. Items from the measure include statements such as “Sometimes I have thoughts of killing myself” and “I am having thoughts of suicide, but these thoughts are somewhat under my control”. In a previous study that examined suicide risk among college students, the scale demonstrated good internal reliability (*α* = 0.89) [50]. The HDSQ-SS scale demonstrated excellent internal consistency (*α* = 0.98) in the current study.

## 3. Results

### 3.1. Correlations

Means, standard deviations, and zero-order correlation coefficients of optimism, hope and its subscales, low hopelessness, grit, and suicide ideation, are presented in Table 1. As predicted, hope and related subscales, optimism, grit, and low hopelessness were all significantly positively associated. Additionally, all positive psychological variables were negatively associated with suicide ideation.

### 3.2. Multiple Regression Analysis Results

A linear multiple regression analysis was conducted to examine the predictive relationship of each positive psychological variable (optimism, hope, low hopelessness, and grit) on suicide ideation in the context of the others. Results are presented in Table 2. Only optimism and grit negatively predicted suicide ideation after considering all predictors in a single model.

### 3.3. Construct Level—Principal Axis Factor Analysis

Six composite scores of optimism, the goals, pathways, and agency subscales of the Hope Scale, low hopelessness, and the Grit Scale were entered in a principal axis factor analysis (PAF), with oblimin rotation, to investigate relationships between these positively associated constructs. All composite scores had communalities above 0.2, indicating that the composite scores shared a substantial amount of variance with the factor they loaded.

The Kaiser–Meyer–Olkin (KMO) sampling adequacy statistic of 0.874 and Bartlett’s test of sphericity were significant (χ^2^ (15) = 1899.29, *p* < 0.001), indicating that the matrix of the six composite scores was sufficiently factorable. The scree plot of the PAF was examined to determine the number of factors rotated to final solution. The scree plot indicated the existence of one large factor and no transitional factors. Similarly, analysis of initial eigenvalues indicated that only one factor contained an initial eigenvalue above 1.0. Factor loadings above 0.40 were considered significant, indicating that the factor that the variable loaded on explained at least 16% of the variance of the variable. All six composite scores loaded on the one factor and explained 59.38% of the variance (optimism λ = 0.70; goals λ = 0.82; pathways λ = 0.69; agency λ = 0.91; low hopelessness λ = 0.81; grit λ = 0.70). Overall, results of the PAF conducted at the construct level suggest that hope, optimism, grit, and low hopelessness have similarities that are characterized by one common element.

### 3.4. Item Level—Principal Axis Factor Analysis

To further investigate whether there were unique relationships at the item level, all items from the LOT-R, HS-R2, BHS, and the Grit Scale were entered in a PAF with oblimin rotation. Fifty-four of the 56 items had an extracted communality above 0.2, indicating that these items shared a substantial amount of variance with the factor(s) that they loaded on. Because of low communality, HS-R2 item 10 (extracted communality = 0.13) and HS-R2 item 17 (extracted communality = 0.12) were withheld from subsequent analyses. The matrix of the remaining 54 items was sufficiently factorable (KMO = 0.945; Bartlett’s test of sphericity (χ^2^ (1540) = 16,600.04, *p* < 0.001)). The scree plot of the PAF was examined to determine the number of factors rotated to final solution. The scree plot indicated one large factor and four transitional factors. Eleven factors had initial eigenvalues above 1.0, but a substantial decrease in both initial eigenvalues and percent of variance explained was seen between factors five and six. Because of these indicators, five factors were rotated to a final solution using oblimin rotation, explaining 46.69% of the variance (see Table 3 for factor loadings of all items). Additional PAFs were conducted in efforts to account for any potential method effects, including a forced four factor solution to examine variance due to the low eigenvalue of the fifth factor, and a forced two factor solution. No significant differences emerged across the PAFs. Data and tables are available upon request.

The structure matrix of the five-factor solution was examined to determine the nature of the factors. This revealed that all 54 items loaded on at least one factor, with most items cross loading on other factors in addition to the factor that explained the most variance of the item. The first factor consisted of 11 items and reflected goal setting and pursuing goals with diligence and dedication (*Stick-to-Itiveness*, e.g., Hope Scale: “There are lots of ways around a problem.”). The second factor consisted of 12 items and reflected negative future expectancies and failures (*Poor Future*, e.g., Beck’s Hopelessness Scale: “I might as well give up because there is nothing I can do about making things better for myself”). The third factor consisted of 6 items and reflected engagement with goals and ability to focus and follow through on each goal (*Consistency of Interest*, e.g., the Grit Scale: “I finish what I begin.”). This term is consistent with the terminology used by Duckworth et al., 2007. The fourth factor consisted of 14 items and reflected positive future expectations and enthusiasm (*Positive Future*, e.g., “Revised Life Orientation Test: In uncertain times, I usually expect the best.”). The fifth factor consisted of 11 items and reflected difficulty in determining ways to achieve goals and inability to persevere through barriers (*Poor Pathways*; Hope Scale: “I have difficulty finding ways to solve problems.”). Table 4 depicts the number and percentage of the hope, optimism, grit, and low hopelessness items that most strongly loaded on each of the retained factors. The five factors shared small to moderate correlations with each other (Table 5). Results of the item-level factor analysis suggest that in addition to the commonalities shared across the constructs and across the items, there are, in fact, unique components of each construct that differentially map onto five distinct factors.

### 3.5. Association of Factors with Suicide Ideation

To determine the relationship between the five extracted factors and suicide ideation, factor scores were saved and correlated with suicide ideation (Table 5). Overall, results indicated that four of the five factors (with the exception of Factor 1, *Stick-to-Itiveness*) were correlated with suicide ideation. Factors 2 (*Poor Future*) and 5 (*Poor Pathways*) were positively correlated with suicide ideation, while Factors 3 (*Consistency of Interest*) and 4 (*Positive Future*) were negatively correlated with suicide ideation. To determine whether there were significant relationships beyond simple associations between the factors and suicide ideation, a linear multiple regression analysis was conducted with the five factor scores entered as predictors of suicide ideation (Table 6). When examined in the context of the other factor scores, Factors 2 (*Poor Future*), 3 (*Consistency of Interest*), and 4 (*Positive Future*) continued to predict suicide ideation.

## 4. Discussion

The current study aimed to examine the common and unique relationships between optimism, hope, and grit, as well as their collective and unique associations with suicide ideation. The approach taken in this study may also help provide a more parsimonious view of protective factors for suicide, as areas of redundancy can be identified and removed while unique protective qualities can be identified and targeted. Relationships between study variables were examined through bivariate correlations, multiple linear regression analyses that predicted suicide ideation, and construct and item-level factor analyses. As hypothesized, all positive psychological variables were positively related to each other, and were differentially predictive of suicide ideation. When examined as simultaneous predictors of suicide ideation, only optimism and grit continued to predict suicide ideation, while hope (and low hopelessness) were no longer related. The item-level factor analysis demonstrated a five-factor solution that revealed the following constructs: (1) *Stick-to-Itiveness*, (2) *Poor Future*, (3) *Consistency of Interest*, (4) *Positive Future*, and (5) *Poor Pathways*. Results demonstrated associations between four of the five factors and suicide ideation. Finally, in the simultaneous prediction of suicide ideation, Factors 2, 3, and 4 continued to predict suicide, while Factors 1 and 5 were no longer related. The overall conclusions, potential implications, and limitations will be discussed.

Previous research examining the protective nature of positive psychology variables on suicide-related outcomes has not examined the differential impact of optimism. While studies have demonstrated the protective qualities of hope on suicide ideation, it may be that other constructs, in this case optimism and grit, are stronger predictors of suicide ideation when examined simultaneously. These findings are not particularly surprising as past research has indicated that goal-directed activity (present in grit) and optimistic thinking styles are generally protective against suicide-related outcomes [15,29]. It may be possible that goal-directed behavior is a distal protective factor of suicide that increases individual’s optimism or positive expectations for the future. Diligently engaging in goal-related pursuits is also likely to increase a person’s probability of actually reaching important goals. The subsequent success toward goals encourage positive expectancies for future success, less suicide ideation, and greater reasons for living.

In the current study the factors of *Poor Future*, *Consistency of Interest*, and *Positive Future* independently predicted suicide ideation. Interestingly, only grit items loaded onto *Consistency of Interest*, which accounted for the most variance in predicting suicide ideation, and five of the six total optimism items loaded onto *Positive Future*. *Poor Future* was composed almost solely of low hopelessness items. It stands out that both grit and optimism were the only measures that predicted suicide ideation in the initial multiple regression analysis, and items from these measures largely composed two of the three final factors that predicted suicide ideation in the final regression analysis. Future studies, specifically focused on positive psychological concepts as protective against suicide related outcomes, should continue to explore the unique factor items indicated in the current study. While these initial findings are preliminary, it may be possible that greater scientific understanding of the uniquely associated measure items and predictive factors could lead to the identification of distinctive protective components against suicide related behaviors.

Many current models of suicide tend to focus on suicide risk, leaving little room to examine suicide from a more comprehensive standpoint. This study of suicidal behavior is multifaceted, and ] = a theoretical model of suicide is necessary to understand such complexity (beyond simple associations between suicide and related variables [51]). A more comprehensive approach that incorporates several protective factors can possibly help to enhance our scientific understanding of suicide resilience within individuals, and/or populations, that are affected by multiple risk factors for suicide, yet do not go on to ideate, attempt, or die by suicide. The ultimate goal of advocating for the creation of a more comprehensive model or theory of suicide—one that is explicitly inclusive of empirically supported protective factors—is to promote an increase in the scientific study of early suicide prophylaxis and resilience building. While the accurate prediction of those who are at near imminent risk of death by suicide is of utmost importance, the diligent structured empirical study of protective factors and resilience could theoretically enhance early prevention efforts, as well as more precise suicide risk assessment.

Current findings may offer several important implications. The common and unique relationships between optimism, hope, low hopelessness, and grit, along with their association with suicide ideation, highlight the distinct qualities that may eventually lead to possible reductions in suicide ideation. Finally, previous research suggests that some of the traits examined here, such as hope [52] and optimism [53], may also be state like. Future research should explore the trait vs. state qualities of hope and optimism both broadly and as they concern suicide-related outcomes, as they may have the capability to be fostered in clients within a therapeutic setting. Future research is also needed to provide further validation of the utility of targeting these constructs within the context of treatment.

It is important to acknowledge the limitations of the current study. The participants in this study were university students, young adults, and white. While participants did endorse suicide ideation and risk factors for suicide, the results of the current study may not generalize to populations at higher risk for suicide (e.g., psychiatric inpatients, people with multiple past suicide attempts, or those in different age or ethnic groups). Additionally, the sample was predominantly female. Previous suicide literature has documented gender differences in the development and maintenance of suicide ideation and behaviors. For example, girls are more likely to have higher risk of depression and post-traumatic stress disorder as a risk factor for suicide [54]. The study design was cross-sectional, prohibiting the ability to make causal inferences. Future research should investigate resilience in more diverse samples and using longitudinal designs.

## Figures and Tables

**Table 1 ijerph-17-09588-t001:** Means, Standard Deviations, and Correlation Coefficients of Optimism, Hope and its Subscales, Low Hopelessness, Grit, and Suicide Ideation.

Variable	1	2	3	4	5	6	7	8
1. Optimism	-							
2. Hope	0.57 **	-						
3. Goals	0.49 **	0.89 **	-					
4. Agency	0.58 **	0.92 **	0.78 **	-				
5. Pathways	0.45 **	0.84 **	0.59 **	0.65 **	-			
6. Low Hopelessness	0.68 **	0.71 **	0.63 **	0.68 **	0.56 **	-		
7. Grit	0.44 **	0.65 **	0.61 **	0.68 **	0.44 **	0.52 **	-	
8. Suicide Ideation	−0.27 **	−0.18 **	−0.11 *	−0.24 **	−0.11 *	−0.25 **	−0.22 **	-
*M*	16.62	113.25	37.71	39.36	36.19	64.53	30.36	0.21
*SD*	4.35	16.23	5.85	6.51	5.95	11.75	6.67	0.93

** *p* < 0.01; * *p* < 0.05.

**Table 2 ijerph-17-09588-t002:** Multiple Regression Analyses of all Positive Psychological Variables and Suicide Ideation.

	Suicide Ideation		
	B	SE	*t*
Optimism	−0.04 *	0.01	−3.34
Hope	0.01	0.00	1.76
Low Hopelessness	−0.01	0.01	−1.86
Grit	−0.02 *	0.01	−2.62

* *p* < 0.01.

**Table 3 ijerph-17-09588-t003:** Factor Loadings and Communalities from a PAF with Oblimin Rotation and Five-Factor Solution for 54 Items.

Item	1	2	3	4	5	h^2^
	Stick-to-Itiveness	Poor Future	Consistency of Interest	Positive Future	Poor Pathways	
Hope1 (Agency)				0.457	**−0.586**	0.512
Hope2 (Goals)	**0.568**	−0.330		0.420	−0.459	0.553
Hope3 (Pathways)	0.364			0.335	**−0.469**	0.446
Hope4 (Goals)	**0.632**	−0.334		0.435	−0.412	0.563
Hope5 (Goals)			0.351		**−0.377**	0.342
Hope6 (Agency)	**0.574**	−0.423		0.451	−0.538	0.564
Hope7 (Pathways)		−0.306		0.424	**−0.760**	0.614
Hope8 (Agency)	0.477	−0.336	0.383	0.470	**−0.711**	0.652
Hope9 (Pathways)	0.327	−0.330		0.438	**−0.728**	0.599
Hope11 (Agency)	0.453	−0.329	0.392	0.444	**−0.632**	0.596
Hope12 (Goals)	**0.542**			0.373	−0.397	0.489
Hope13 (Goals)	0.449	**−0.548**		0.438	−0.441	0.532
Hope14 (Pathways)		−0.422		0.523	**−0.711**	0.609
Hope15 (Agency)	**0.533**	−0.448		0.431	−0.324	0.471
Hope16 (Goals)	0.464	−0.440		0.493	**−0.622**	0.569
Hope18 (Agency)	**0.564**	−0.479	0.396	0.523	−0.555	0.673
LOT-R1		−0.313		**0.563**	−0.380	0.484
LOT-R3		−0.315		0.380	**−0.456**	0.434
LOT-R4		−0.379		**0.642**	−0.380	0.544
LOT-R7		−0.479	0.302	**0.518**	−0.513	0.570
LOT-R9		−0.494	0.310	**0.555**	−0.487	0.572
LOT-R10		−0.512		**0.650**	−0.436	0.559
GRIT1	**0.560**	−0.344		0.423	−0.346	0.442
GRIT2			**0.526**			0.311
GRIT3			**0.643**			0.468
GRIT4	0.388			0.422	**−0.423**	0.390
GRIT5			**0.550**			0.360
GRIT6	**0.657**	−0.429		0.429	−0.395	0.574
GRIT7			**0.685**			0.520
GRIT8	0.356		**0.648**	0.325	−0.385	0.525
GRIT9	**0.520**		0.392	0.336		0.485
GRIT10	**0.577**			0.338	−0.323	0.474
GRIT11			**0.545**			0.418
GRIT12	**0.606**	−0.355		0.455	−0.389	0.536
BHS1	0.396	−0.518		**0.750**	−0.524	0.692
BHS2	0.303	**−0.730**		0.394	−0.325	0.586
BHS3		−0.314		**0.637**	−0.338	0.517
BHS4		**−0.368**	−0.320	0.363		0.417
BHS5				**0.585**	−0.369	0.480
BHS6	0.384	−0.390		**0.656**	−0.403	0.563
BHS7		**−0.673**		0.455		0.558
BHS8				**0.589**	−0.347	0.495
BHS9		**−0.605**				0.465
BHS10	0.433	−0.333		**0.627**	−0.483	0.547
BHS11		**−0.789**		0.409		0.658
BHS12		**−0.711**		0.458	−0.396	0.619
BHS13				**0.317**		0.387
BHS14		**−0.747**		0.427	−0.343	0.573
BHS15	0.403	−0.422		**0.930**	−0.510	0.878
BHS16		**−0.848**		0.333	−0.346	0.776
BHS17		**−0.832**		0.357	−0.361	0.732
BHS18		**−0.543**	0.361	0.463		0.542
BHS19	0.347	−0.451		**0.718**	−0.446	0.609
BHS20		**−0.843**		0.395	−0.307	0.759
Initial Eigenvalues	17.022	3.497	2.957	2.490	1.789	-
Extraction Eigenvalues	16.530	3.042	2.373	1.968	1.298	-
Initial % of Variance	31.522	6.476	5.476	4.611	3.313	51.398
Extracted % of Variance	30.611	5.634	4.395	3.645	2.404	46.689

Note: N = 542. Loadings above 0.40 are significant. In the case of cross loadings, bolded numbers represent the strongest loading factors. Factor loadings < 0.30 are suppressed. Hope items are from the Hope Revised Scale and are listed according to subscale: goals, agency, and pathways. LOT-R = optimism, items are from the Revised Life Orientation Test. GRIT = grit, items are from the Grit Scale. BHS = low hopelessness, items are from Beck’s Hopelessness Scale.

**Table 4 ijerph-17-09588-t004:** Number and Percentage of Items That Most Strongly Loaded on Each Retained Factor.

	1	2	3	4	5
Optimism	0	0	0	5	1
	(0%) ^a^	(0%) ^a^	(0%) ^a^	(35.7%) ^a^	(9.1%) ^a^
	(0%) ^b^	(0%) ^b^	(0%) ^b^	(83.33%) ^b^	(16.67%) ^b^
Hope *	6	1	0	0	9
	(54.5%) ^a^	(8.33%) ^a^	(0%) ^a^	(0%) ^a^	(81.8%) ^a^
	(37.5%) ^b^	(6.25%) ^b^	(0%) ^b^	(0%) ^b^	(56.25%) ^b^
Low Hopelessness	0	11	0	9	0
	(0%) ^a^	(91.67%) ^a^	(0%) ^a^	(64.3%) ^a^	(0%) ^a^
	(0%) ^b^	(55%) ^b^	(0%) ^b^	(45%) ^b^	(0%) ^b^
Grit	5	0	6	0	1
	(45.5%) ^a^	(0%) ^a^	(100%) ^a^	(0%) ^a^	(9.1%) ^a^
	(41.67%) ^b^	(0%) ^b^	(50%) ^b^	(0%) ^b^	(8.33%) ^b^

Note: ^a^ Denotes percentage of items on each retained factor composed by study measures. ^b^ Denotes percentage of items on each study measure that most strongly loaded on the retained factors. Percentage values = percent of the study measure that loaded on the retained factor. * Two items of the pathways subscale of the Hope Scale (items 10 and 17) demonstrated low communality in the set and did not load on any of the five factors.

**Table 5 ijerph-17-09588-t005:** Correlation Coefficients, Means, and Standard Deviations of the Five Retained Factor Scores and Predictors of Suicide Ideation.

Variable	1	2	3	4	5	6
1. *Stick-to-Itiveness*	-					
2. *Poor Future*	−0.28 **	-				
3. *Consistency of Interest*	0.15 **	−0.24 **	-			
4. *Positive Future*	0.40 **	−0.50 **	0.25 **	-		
5. *Poor Pathways*	−0.40 **	0.43 **	−0.32 **	−0.61 **	-	
6. Suicide Ideation	−0.05	0.24 **	−0.25 **	−0.24 **	0.17 **	-

Note: Variables in italics are retained factors from the item-level factor analysis. ** *p* < 0.01.

**Table 6 ijerph-17-09588-t006:** Multiple Regression Analysis of the Five Retained Factors and Suicide Ideation.

	Suicide Ideation		
	B	SE	*t*
*Stick-to-Itiveness*	0.08	0.05	1.69
*Poor Future*	0.13 *	0.05	2.79
*Consistency of Interest*	−0.19 **	0.05	−4.30
*Positive Future*	−0.15 *	0.05	−2.86
*Poor Pathways*	−0.02	0.06	−0.30

Note: Variables in italics are retained factors from the item-level factor analysis. ** *p* < 0.001; * *p* < 0.01.

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
