# Peer review of "The Common Factors of Grit, Hope, and Optimism Differentially Influence Suicide Resilience"

_ijerph, 2020, doi:10.3390/ijerph17249588_

Round 1

Reviewer 1 Report

Thank you, authors have addressed my comments and concerns.

Reviewer 2 Report

Thanks to the authors for carefully considering the suggested edits and revising the manuscript to reflect relevant changes. The changes have indeed addressed each of my comments (reviewer 2) and have resulted in a much more readable, concise, and improved manuscript. 

This manuscript is a resubmission of an earlier submission. The following is a list of the peer review reports and author responses from that submission.

Round 1

Reviewer 1 Report

Thank you for an opportunity to review this manuscript presenting a study on how the common factors of grit, hope, and optimism differentially influence suicide resilience. It's an innovative study looking at relationships between positive psychology variables and suicidal ideation in university students in the USA. Several suggestions to improve the quality of the mss:
1. Title - please define 'suicide resilience'. The text of the mss does not include references to 'suicide resilience', it talks about 'protective factors' instead. I suggest being consistent in the language used.
2. Abstract - please, double check lines 22-23 as they seem inconsistent with lines 291-292.
3. Introduction - I suggest streamlining and shortening this part of the mss as there are repetitions. In lines 52-53, please explain meaning of 'simply because they are positively valenced'. The Introduction ends with several hypotheses; however, Results and Discussion do not refer to these. Again, I recommend being consistent, e.g., structuring the mss around the hypotheses proposed.
4. Materials and Methods - I suggest creating a separate subsection on statistical methods, e.g., including lines 132-138. Can the authors quote the items on Hopelessness-Depression Symptom Questionnaire – Suicidality Subscale (HDSQ-SS) which measure suicide riks?
5. Results - I wonder if some of the statistical outcomes could be presented as an onine Appendix?
6. Discussion/Conclusions - I suggest merging these two sections as currently Discussion is a brief repetition of the main study results only. I also suggest creating a separate sections on Study Implications (currently line 359 and onwards). I recommend caution and not including 'clinical recommendations' as such (e.g., management of 'patients' - line 365) as the current study presents only preliminary data obtained is a sample of Uni students in the USA. Similarly, I recommend caution in regard to overgeneralising the study findings and conclusions related to 'suicide' - the current study looked at suicidal ideation only, which was measured using 4 items of HDSQ-SS.
7. I also suggest creating a separate section on study limitations (currently line 379 and onwards).

Reviewer 2 Report

Thank you for submitting this manuscript for review. The search for protective and positive psychology variables associated with suicide risk is an important endeavor and one that has been taken on by numerous studies.

This manuscript needs considerable work before it will be ready for publication in any outlet. 

This paper reads more like a data analysis project rather than a well-conceptualized, intentional study for larger implications or knowledge-building. It does not necessarily contribute anything substantial to what is already known in the larger body of literature. 

The first and most pressing of concerns is the readability of this manuscript. There are considerable grammar mistakes throughout the paper, beginning from line 1, for example, as the word data is plural. Lines 37-40 all have writing errors, there is a redundancy in use of phrasing, and well as a confusing use of terms that have not yet been defined or well explained.

Introduction: The introduction has quite a bit of redundancy without clarity or specificity. For example, you do not explain, at all, the phenomenon of suicide, outside of some stats on current rates. You provide a list of your variables, but it is conceptually murky regarding your justification for the need for this study. In other words, you do not provide a strong argument about why this is filling a gap in the literature. You end the introduction pointing to a study that was "left hanging". There was no real transition there.

Methods: You need a demo table. What was the SD of age?

The setting, recruitment, and sampling are seriously underdeveloped sections. This was obviously a convenience sample, yet, you do not mention this. You do not mention what kind of class this was- a psychology course? Sociology? Nursing? This matters. When was it done in the semester? This was a cross-sectional study, so had these students been exposed to some learning on the subject matter? 

Your measures descriptions do not include previous findings in a similar sample; it is important to include their reliability and validity so your audience knows that these are measures shown to hold up in prior research.

Your discussion was more of a narrative summary of your results instead of a discussion. Your conclusion section had a bit more discussion there, but your conclusions do not align with your results. You overstate the utility of your findings. One very valuable statement you make is that more research should be done on the "trait versus state" of these variables. This is very important. The grit literature from Duckworth suggests that grit is a trait.

Another underdeveloped section is the limitations. There are numerous other limitations. For example, over three-quarters of your sample is female. The literature is clear about the differences in gender and the experience of suicidality from thoughts to attempts to death. Also, I wonder about the differences in grit, hope, and other positive psychological factors related to gender. 

Suicide is a very complex phenomenon. Personal characteristics, including the existence of protective factors and positive psychological traits, may have a relationship to the spectrum of the suicide experience, but this paper is not nuanced that way. There are assertions that these things have an actual impact on suicidality, but this is not a conclusion that can be reached in this sample or with these methods.